# Kinect-Based Gait Analysis System Design and Concurrent Validity in Persons with Anterolateral Shoulder Pain Syndrome, Results from a Pilot Study

**DOI:** 10.3390/s24196351

**Published:** 2024-09-30

**Authors:** Fredy Bernal, Veronique Feipel, Mauricio Plaza

**Affiliations:** 1Faculty of Engineering, Universidad Militar Nueva Granada, Bogotá 110111, Colombia; 2Department of Functional Anatomy, University Libre de Bruxelles, 1170 Brussels, Belgium; veronique.feipel@ulb.be; 3Faculty of Medicine, Universidad Militar Nueva Granada, Bogotá 110231, Colombia; mauricio.plaza@unimilitar.edu.co

**Keywords:** Kinect-based motion capture, gait asymmetries, shoulder injuries, rotator cuff syndrome, repeated measures ANOVA, statistical parametric mapping SPM

## Abstract

As part of an investigation to detect asymmetries in gait patterns in persons with shoulder injuries, the goal of the present study was to design and validate a Kinect-based motion capture system that would enable the extraction of joint kinematics curves during gait and to compare them with the data obtained through a commercial motion capture system. The study included eight male and two female participants, all diagnosed with anterolateral shoulder pain syndrome in their right upper extremity with a minimum 18 months of disorder evolution. The participants had an average age of 31.8 ± 9.8 years, a height of 173 ± 18 cm, and a weight of 81 ± 15 kg. The gait kinematics were sampled simultaneously with the new system and the Clinical 3DMA system. Shoulder, elbow, hip, and knee kinematics were compared between systems for the pathological and non-pathological sides using repeated measures ANOVA and 1D statistical parametric mapping. For most variables, no significant difference was found between systems. Evidence of a significant difference between the newly developed system and the commercial system was found for knee flexion–extension (*p* < 0.004, between 60 and 80% of the gait cycle), and for shoulder abduction–adduction. The good concurrent validity of the new Kinect-based motion analysis system found in this study opens promising perspectives for clinical motion tracking using an affordable and simple system.

## 1. Introduction

Human movements can be described as a well-coordinated set of interactions of the musculoskeletal system (bones, muscles, ligaments, and joints) [1,2]. The injuries or damages of any individual element of this system can generate a degradation of balance and stability or changes in the mechanical behavior of the general system, altering movements of the damaged structure and/or of other joints, even those not directly related [3]. The analysis of the human gait consists of obtaining periodic parameters of movement of the upper and lower extremities, which reflect individual patterns. When presenting disorders or changes in some structures of the musculoskeletal system, global changes in other body segments related to gait are possible, according to [4]. Using motion capture system analysis, it is possible to quantify the changes generated by such alterations throughout the human body [5].

Currently, pathologies, injuries or alterations of the shoulder occur in Colombia in 78 per thousand inhabitants. According to Federación de aseguradoras Colombianos (FASECOLDA) [6] 28% of the diagnoses are of occupational origin, the most common being rotator cuff syndrome with 35% of these cases [7,8]. It could therefore be relevant to carry out a study monitoring, identifying, and characterizing the motion parameters related to different shoulder disorders and how these alter the gait balance in persons who present this type of disease, specifically rotator cuff syndrome.

Human gait analysis uses systems that enable the measuring of the global movement of the human body. These systems, known as motion capture systems, make use of different technologies to obtain data, the most common being optical and inertial systems [9]. Optical systems use sensors or cameras that detect estimated anatomical positions. Through recurrent calculations and data extraction algorithms, an estimate of joint movements and spatiotemporal parameters of interest can be obtained [10].

The objective of the research was to develop a motion capture system aimed at reducing costs and difficulty of implementation, enabling the ability to obtain measurements with sufficient precision compared to traditional systems endorsed for commercial use. For this reason, a search was carried out for technologies that would allow for the capture of information and would enable the obtainment and estimation of the pertinent parameters. Based on the search for technologies, the design and implementation of the motion capture system were carried out for its validation, using a commercial device as a reference to determine if there are significant differences between the systems used.

## 2. Materials and Methods

### 2.1. Subjects

**To adhere to government regulations, a protocol for data collection and management was implemented and approved by an ethics committee, including the creation of an informed consent form. This form, in accordance with the Declaration of Helsinki and Resolution No. 8430 of 1993 from the Ministry of Health in Colombia, informed volunteers about the type of test, associated risks, and patient rights during the measurements**. With the aim of fulfilling the study’s objective, a search was conducted to recruit volunteers for test participation. During the selection process, specific parameters and conditions were established and considered for potential volunteers.

### 2.2. Inclusion Criteria

−Diagnosis of rotator cuff syndrome or anterolateral shoulder pain syndrome performed by a medical professional of the Colombian healthcare systems;−Evolution of the disorder for 18 months as minimum;−Women and men aged between 20 and 45 years.

In this study, no proof or medical certificates of diagnosis were requested to ensure patient confidentiality and the protection of personal data.

### 2.3. Exclusion Criteria

− Other types of alterations diagnosed by orthopedists or health professionals that may affect the motor behavior of the subject.

The selected sample consisted of eight men and two women whose environment primarily involved office or low physical effort jobs. All participants were located in the city of Bogotá, Colombia, all with rotator cuff syndrome pathology in the right extremity, with a mean age of 31.8 ± 9.8 years, height of 173 ± 18 cm, and a weight of 81 ± 5 kg. Participants were asked to report their pain levels (maximum and average pain experienced throughout the evolution of the disorder as well as present pain on the test day) on a numeric scale of 0–10, where 0 was equivalent to no pain and 10 to the maximum bearable pain; the obtained results are shown in Table 1.

### 2.4. Instrumentation

As a reference for movement capture, the Clinical 3DMA^®^ system SST SYSTEMS [11] was used, in a configuration of 8 Optitrack infrared cameras with a resolution of 640 × 480 and a sampling frequency of 100 FPS. This system makes use of reflective markers located at specific points on the body surface, allowing the estimation of angular movements of the limbs and joints. For this study, the whole-body protocol was used, which includes a total of 19 markers (Figure 1) for the simultaneous tracking of angular and spatiotemporal parameters of the lower and upper limbs. Markers were placed by the same trained researcher in all cases to limit the risk of inaccuracies.

To use the parameters captured by the Clinical 3DMA, it was necessary to extract goniometric curves for each joint throughout the entire test. However, this extraction process needs to be performed manually for each desired joint and movement to be analyzed. Consequently, a pre-processing stage becomes essential to consolidate the acquired data before it can be utilized in the conducted study.

On other hand, the motion capture system developed in this study uses the Microsoft^®^ Kinect V2, one of the most advanced and well-known commercial motion sensing equipment, with the following characteristics [12]:−70° horizontal and 60° vertical field of view;−1920 × 1080 P camera resolution;−Sensor depth range 0.5–4.5 m;−USB 3.0 interface;−Sampling frequency 30 Hz.

A motion capture and analysis software were developed in MATLAB^®^ 2021a environment, with additional add-ons (MATLAB support package for USB webcams, Kinect for Windows Sensor Imaging Toolbox Support Pack). These additional add-ons allow the extraction of information directly from the Kinect, specifically the spatial positions of predefined joint markers. For this study, the following markers (Table 2) were used, as they are markers located on relevant joints for gait analysis.

Developments designed in MATLAB can be conceived to be exported as a system independent from the base code. Considering the computational resources necessary for Kinect V2 operation and the premise of system portability, it was proposed to develop the capture software separately from the analysis system.

The motion capture software comprises a patient identification module (Figure 2). The capture software was developed following the flow diagram shown in Figure 3.

The developed software has three main functionalities.

Capture: Allowing the start and stop of motion capture. During capture, a signal indicates that motion capture is in progress, but without real time display.

Visualization: This function allows users to assess the success of the data capture in a straightforward manner. The captured data is presented in an additional window, allowing for visual inspection. As the test progresses and the entire body movement is reproduced, a visual witness indicates the complete reproduction of the movement. Once the test is concluded, the witness deactivates, granting the option to close the display and securely save the data obtained from the test. If the software user identifies any error in the capture during the visualization, it can be redone.

Save test: This feature facilitates the storage of test data in three distinct files. The first file includes personal data and test conditions, which are exported as a text document. The second file comprises the exported data presenting the three-dimensional coordinates of the joints throughout the entire test. Lastly, the third file contains the collected data in a MATLAB variable format, allowing a seamless integration with analysis software for further processing and in-depth data analysis.

The developed software identified the initial and final positions of each trial by utilizing markers placed on both ankles and the base of the spine. By calculating the angles of rotation and elevation using matrix rotations, any variations in orientation were effectively compensated. This rotation process is necessary to align the coordinate axes of the capture system with the global coordinate axes of the user, thus avoiding calculation errors when estimating body planes during data extraction. Additionally, the mirror effect was accounted for in this process; the compensation of the mirror effect was carried out, which when viewing the images inverts the right and left sides.

Joint rotations were estimated using the markers adjacent to the corresponding joints, for example, for knee flexion–extension, it was necessary to take the coordinates of the hip, knee, and ankle markers. With vectors created between these markers and by calculating the angle between them, joint rotation can be estimated by Equation (1).
(1)$$\cos\theta =\frac{\overset{⃑}{u}\times \overset{⃑}{v}}{\left|\overset{⃑}{u}\right|\times \left|\overset{⃑}{v}\right|}$$
where ***θ*** is the generic name given to the calculated angle, it is only used in the equation as an informative element. $\overset{⃑}{u}\mathrm{and}\overset{⃑}{v}$ are the vectors created between the joint and the complementary markers, as an example $\overset{⃑}{u}$ = vector between positions of the knee and the ankle markers. This calculation is necessary for each frame captured, equivalent to 30 data per second of capture.

It should be noted that this calculation does not enable the estimation of the 3D rotation values (due to the lack of additional markers it is not possible to identify each body segment as a plane but as a vector) of the joints and, given that, the joint movement is not limited to a single degree of rotational freedom. Thus, the final angle obtained from the calculation is only an approximation (Figure 4).

For variables such as shoulder and hip flexion–extension, taking into account that the goniometry values are calculated in relation to the anatomical planes, the markers located on the spine are used, and projections of these are created in comparison with the articulation of the movement. For the flexion–extension movement of the shoulder, Spine_shouder and Spine_mid were used as additional markers. With these, and by creating a vector with the beginning of the shoulder to be evaluated, an equivalent to the sagittal plane is created.

Likewise, for the hip flexion and extension, an additional vector is constructed emulating the sagittal plane using the Spine_base and Spine_mid markers, to define virtual markers located proximal to the joints to be analyzed. Likewise, for the shoulder and hip flexion–extension with respect to the vertical (elevation), a vertical vector was created using the vertical axes to calculate the elevation angle.

Shoulder and hip flexion–extension to vertical, according to [13], can be considered as the elevation of the limb with respect to the gravitational line and not towards the coronal plane.

Although the Clinical 3DMA Capture System enables the extraction of other parameters (pelvic rotation, knee adduction and abduction or rotation, or ankle flexion–extension), these could not be obtained by the Kinect system due to intrinsic limitations, such as noise in the measurements.

In total it is possible to obtain eight parameters:−Shoulder flexion–extension (Shoulder FE);−Arm elevation (Shoulder flexion–extension with respect to vertical);−Shoulder abduction–adduction (Shoulder AA);−Elbow flexion–extension (Elbow FE);−Hip flexion–extension (Hip FE);−Thigh elevation (Hip flexion–extension with respect to vertical);−Hip abduction–adduction (Hip AA);−Knee flexion–extension (Knee FE).

Moreover, gait cycle events (heel strike and toe off) were estimated by analyzing the vertical displacements of markers, such as those placed on the ankles. Furthermore, after identifying the specific gait cycle to be analyzed, the developed software allows for a preview of the cycle. If required, adjustments to the start and end times can be made manually by the user. Once the start and end times are confirmed, the software proceeds to normalize the gait cycle, converting the values to a standardized range of 0 to 100% (Figure 5).

Test protocol:

To reduce the variables that could affect the results of the tests, volunteers were invited to follow recommendations and indications on the day assigned to the test:Wear comfortable or sports clothing;Arrive at least one hour before the test;No physical activity on the day of the test;Avoid means of transport that require physical effort.

These pre-test recommendations were given to volunteers to minimize potential variables affecting test results, with the intention of ruling out alterations caused by physical fatigue, discomfort in data collection due to clothing, time constraints, or haste during the test, and to ensure the correct positioning of the reflective markers of the guidance system.

Participants walked in a laboratory setting allowing simultaneous capture of gait data with the Kincapsys and Clinical 3DMA.

During the test execution, a simultaneous capture approach was employed, using both systems. The volunteers were instructed to freely start to walk within the capture space of both the Kincapsys and the STT system. Each patient underwent a total of six trials. To ensure data consistency and minimize potential confounding factors arising from the spatial limitations of the constrained movement capture system, the volunteers were asked to follow the following guidelines: to begin each walking sequence three times with each extremity, and complete a minimum of three steps per trial.

Once the curves were extracted, the results obtained using the different capture systems were compared separately for the affected and the non-pathological sides, considering potential differences between sides due to the nature of the study sample (shoulder injury). To carry out this comparison between measurement systems, two statistical analyses were implemented, the first using the 1D Statistical Parametric Mapping (SPM 1D). [14,15,16]. Through this methodology, it becomes feasible to perform topological inference (instead of time point tests) in order to compare time-series data (such as kinematics curves) by computing a time-series of the Hotelling t statistics. In the context of SPM, t* functions as a threshold to identify statistically significant activations or differences. A second analysis used a repeated measures ANOVA (ANOVA RM) [17,18,19]. An alpha of 5% was used as a significance threshold.

To simplify the comparison using repeated measures ANOVA, angular values were extracted at regular intervals of 10% throughout the gait cycle. In the event of a significant effect detected by the ANOVA, a Tukey post hoc test was conducted to identify specific phases of the gait cycle where discrepancies between the systems were observed. This analytical approach allowed for a comprehensive assessment of any variations in the gait cycle phase across the different systems under investigation.

## 3. Results

Once the tests were performed, the following results were obtained, separated between the non-pathological side and the pathological side. Table 3 and Figure 6 show the results for non-pathological sides.

The comparison between Kinect and Clinical 3DMA on the non-pathological side showed that only two movements displayed a statistically significant difference: shoulder abduction and adduction (at the middle of the gait cycle, *p* = 0.0149), and knee flexion–extension (*p* = 0.0013, between 60 and 80% of the gait cycle) (Figure 6c,h).

According to the work of [14], for comparisons of SPM in 1D1D, it is possible to parameterize the t* as a single value for the region of interest (ROI) of the data. In their work he shows the theoretical consistency between the 0D (point to point) and 1D (curve with single t* value) values with very little variation. The results shown in Table 4 ANOVA RM for the non-pathological side. The analysis shows that the only significant differences found were for shoulder abduction–adduction (*p* = 0.033) and for knee flexion and extension (*p* = 0.046).

For shoulder adduction–abduction and knee flexion–extension, which showed significant differences, the Tukey post hoc test was computed (Table 5).

With the results obtained, the significant difference between the two systems for knee flexion and extension in non-pathological sides was between 60 and 80% of the gait cycle, corresponding to the same result obtained by the SPM 1D method. Similarly, the ANOVA RP analysis revealed a significant difference around 40% of the gait cycle just like the SPM 1D for shoulder abduction and adduction. Furthermore, the comparison of the pathological sides yielded the following results for SPM (Figure 7 and Table 6).

It is possible to see that there are significant differences in two movements: the hip flexion–extension with vertical (elevation), with a *p*-value of 0.0154 (Figure 7d) and knee flexion–extension with a *p*-value of 0.0032 (Figure 7h).

The Table 7 shows the results of the ANOVA RP, indicating significant differences in knee flexion–extension and shoulder flexion–extension.

The Tukey HSD post hoc test for knee flexion–extension and shoulder flexion–extension (Table 8) shows the portions of the gait cycle displaying significant differences for shoulder and knee flexion. These results are consistent with the SPM 1D analysis for the shoulder flexion–extension (Figure 7a), indicating differences between systems at heel strike at the beginning and between 70 and 90% of the gait cycle. For knee flexion–extension, significant differences were found between 60 and 80% of the gait cycle (Figure 7h).

The findings of statistical comparisons between systems are summarized in Table 9. The table highlights movements showing significant differences and specifies the gait cycle phase where these differences were observed.

Since the study did not aim to assess the evolution or behavior of the shoulder injury, it is not possible to provide detailed information on the following aspects: description of damages; treatment compliance (acceptability); and the handling of missing data.

## 4. Discussion

The analysis of the obtained results confirms a significant difference between the measurement systems, particularly for knee flexion–extension movement. This difference is observed in both the pathological and non-pathological sides, encompassing between 60% to 80% of the gait cycle. A systematically lower knee flexion was estimated by the Kinect system, with a mean difference exceeding 15°. This discrepancy can be attributed to the inherent accuracy limitations of the Kinect system, which estimates marker distances using a point mesh methodology. The dependency of the system on the depth information for location estimation of the relevant markers contributes to the increased error observed.

When examining additional movements, notable differences are observed in shoulder abduction and adduction. These disparities are substantiated by SPM and ANOVA RP analyses, particularly on the non-pathological side at around 40% of the gait cycle.

For the pathological side, a significant difference was found in shoulder flexion–extension through the ANOVA RM analysis. However, this difference is not visible through the 1D SPM analysis. This discrepancy may be due to the approximation errors made by the SPM in finding a single valid t* value for 100% of the data. Nevertheless, observing Figure 7a, it appears that the curves of both systems are within the acceptable range of difference according to this method.

In contrast, the thigh elevation in the test conducted on the pathological side reveals a disparity at the onset of the gait cycle, as indicated by the SPM 1D analysis. This discrepancy can be attributed to the challenges associated with marker placement on the hip, which can introduce errors in position estimation by the Kinect system. The observed differences can be attributed to the methodologies employed by the Kinect system for estimating the positions of joint markers, which rely on depth camera and image recognition. In contrast, the reference system, Clinical 3DMA, utilizes multiple cameras and reflective markers to employ position triangulation to achieve a more precise calculation of the actual joint positions, including the corresponding functional planes.

Other low-cost systems such as the one implemented by [20], used measurements with inertial sensors and, when compared to an Optitrack optoelectronic system, showed errors of less than 5° in the obtained results. However, this required the use of eleven sensors placed only on the lower body plus one on the spine. In contrast, this system, compared to the one implemented in this study, demands a more comprehensive patient preparation. On the other hand, in comparison to what was implemented by [21], where their study employed three Kinect devices simultaneously to reduce the observed error, the data calculated by our own development exhibits very close resemblances to the data yielded by the traditional optoelectronic system used as a reference, which used a single sensor. Moreover, taking into consideration the study conducted by [22], it is suggested that the use of the developed application for estimating independent joint movements is feasible.

When considering other technologies for the identification of joint movements, such as that presented by [23], where the use of a smartphone camera allows for the identification of ankle, knee, and hip flexion–extension in the participants of their study, it is important to note that, despite their benefits, these technologies have certain flaws. For instance, the measurement of movements on only one side of the volunteer at a time is affected by the camera’s parallel placement with the runway where the volunteers perform their movements. In contrast, the system developed in this research allows for the simultaneous measurement of movements on both sides of the body. Furthermore, by incorporating depth sensor technology, it enables the calculation of movements that, with traditional 2D technologies, would only be estimated with less precision.

The main advantage of using a motion capture system based on Microsoft’s Kinect V2 lies in the simplification of installation, setup, and costs. The developed system has a starting cost of $350, depending on the chosen computer equipment, plus the value of the developed software, which according to the authors’ approach, is intended to be around $600. In comparison, commercially used systems for the same purpose often have prices exceeding $10,000 for the most affordable ones.

Likewise, the system has a great advantage in the follow-up of the recovery of patients, allowing orthopedic rehabilitation to be supported. This is because although it presents differences with respect to commercial systems, it presents low levels of error between measurements of movements with respect to the same system.

The main limitation of the study lies in the small sample size, primarily linked to the period of data collection. The COVID-19 pandemic resulted in a marked reduction in participants’ willingness to engage in the study due to restrictions, health concerns, and shifts in priorities during this exceptional period. Further studies are needed to confirm and increase the robustness of the preliminary data presented in this work.

## 5. Conclusions

Based on the obtained results, it can be concluded that for the majority of movements, there is no significant difference between the developed motion capture system and the commercial system, with the exceptions of knee flexion–extension, shoulder abduction and adduction, and hip flexion–extension with respect to the vertical (elevation). This finding suggests that the system developed for this specific type of measurement is not only viable but also comparable to commercial systems.

The implemented system, although it shows significant differences in certain movements compared to traditional optoelectronic systems, stands out in its main advantages such as its low cost (less than $400), equipment portability, as only a computer and the Kinect V2 system are required, and user-friendliness. Since it does not require reflective markers for joint identification, it significantly reduces the preparation time for measurements as well as the need for additional supplies and consumables.

## Figures and Tables

**Figure 1 sensors-24-06351-f001:**
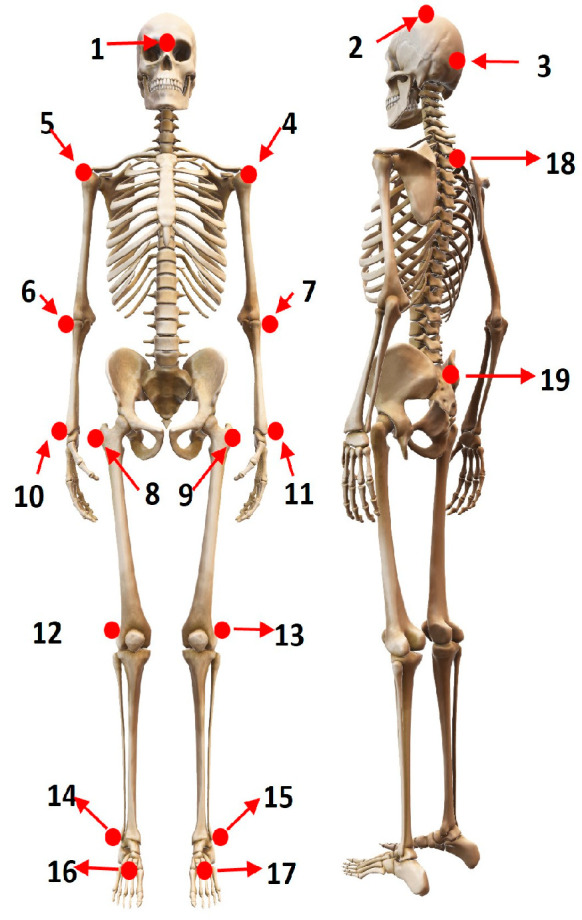
Location of reflective markers for Clinical 3DMA system [11]: 1. Front of the head, 2. Top of the head, 3. Back of the head, 4. Left acromion, 5. Right acromion, 6. Left lateral humeral epicondyle, 7. Right lateral humeral epicondyle, 8. Right trochanter, 9. Left trochanter, 10. Right ulnar styloid process, 11. Left ulnar styloid process, 12. Right lateral femoral epicondyle, 13. Left lateral femoral epicondyle, 14. Right lateral malleolus, 15. Left lateral malleolus, 16. Right 2nd metatarsophalangeal joint, 17. Left 2nd metatarsophalangeal joint, 18. C7 vertebra, 19. Upper part of the sacrum.

**Figure 2 sensors-24-06351-f002:**
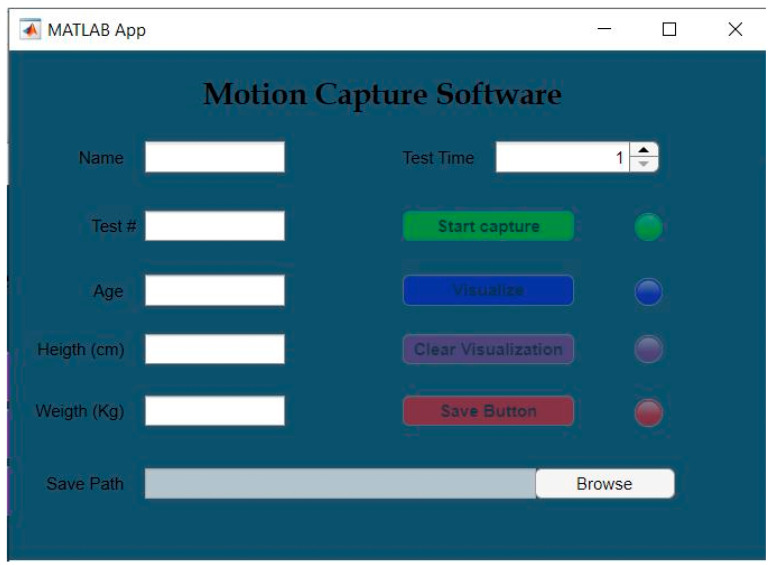
Kinect-based system—motion capture system.

**Figure 3 sensors-24-06351-f003:**
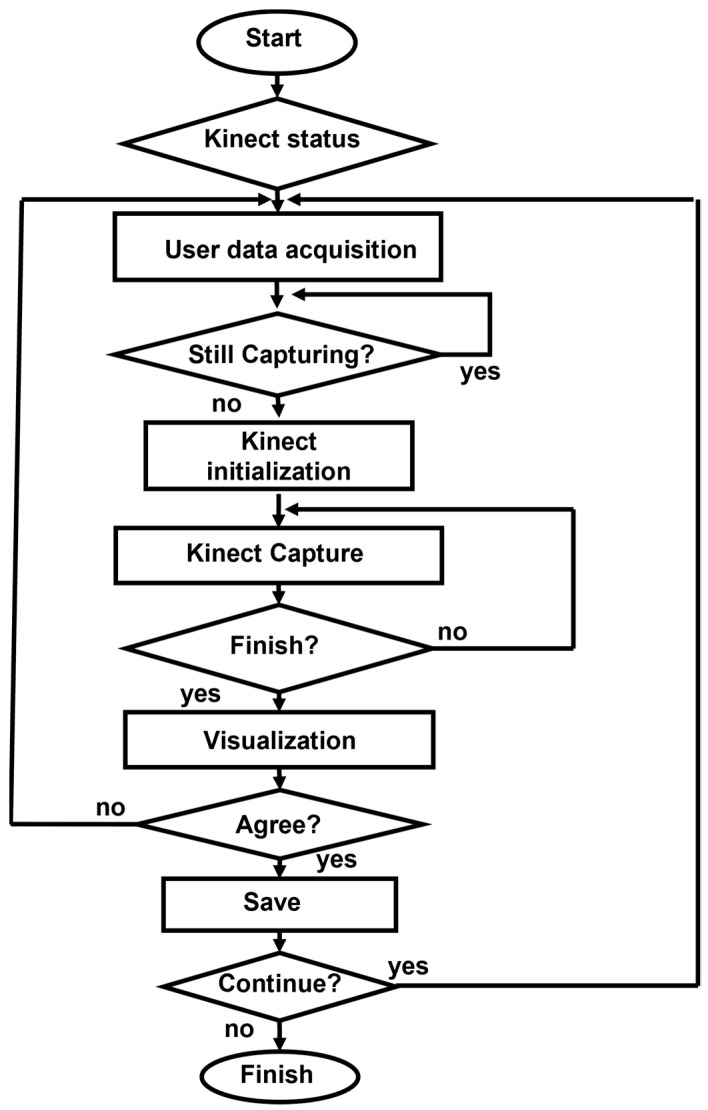
Basic flow diagram of Kinect-based system.

**Figure 4 sensors-24-06351-f004:**
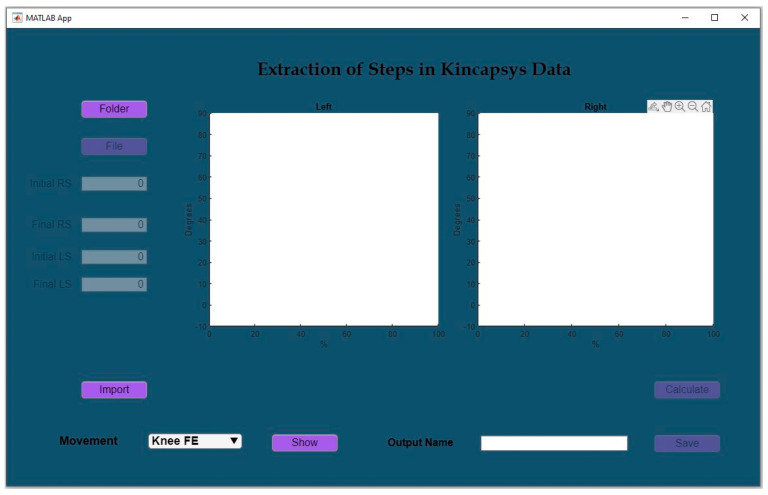
Kinect based system—Data Extraction Software 1.0.

**Figure 5 sensors-24-06351-f005:**
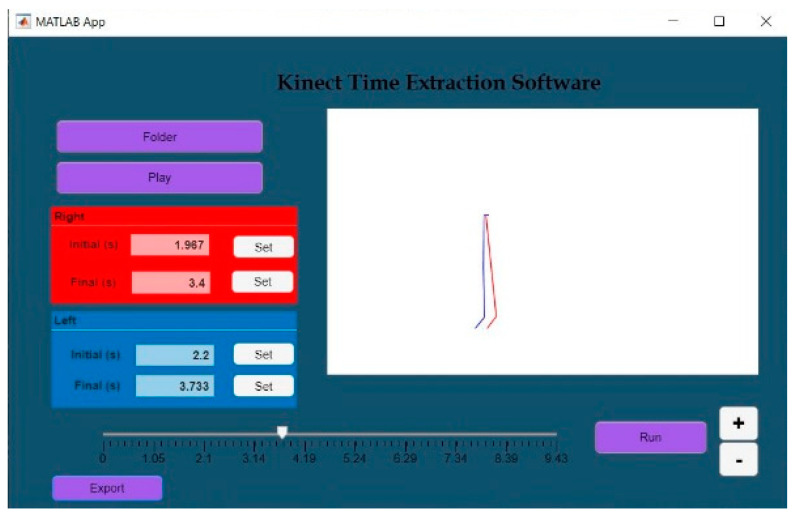
Kinect based system-Time Extraction Software.

**Figure 6 sensors-24-06351-f006:**
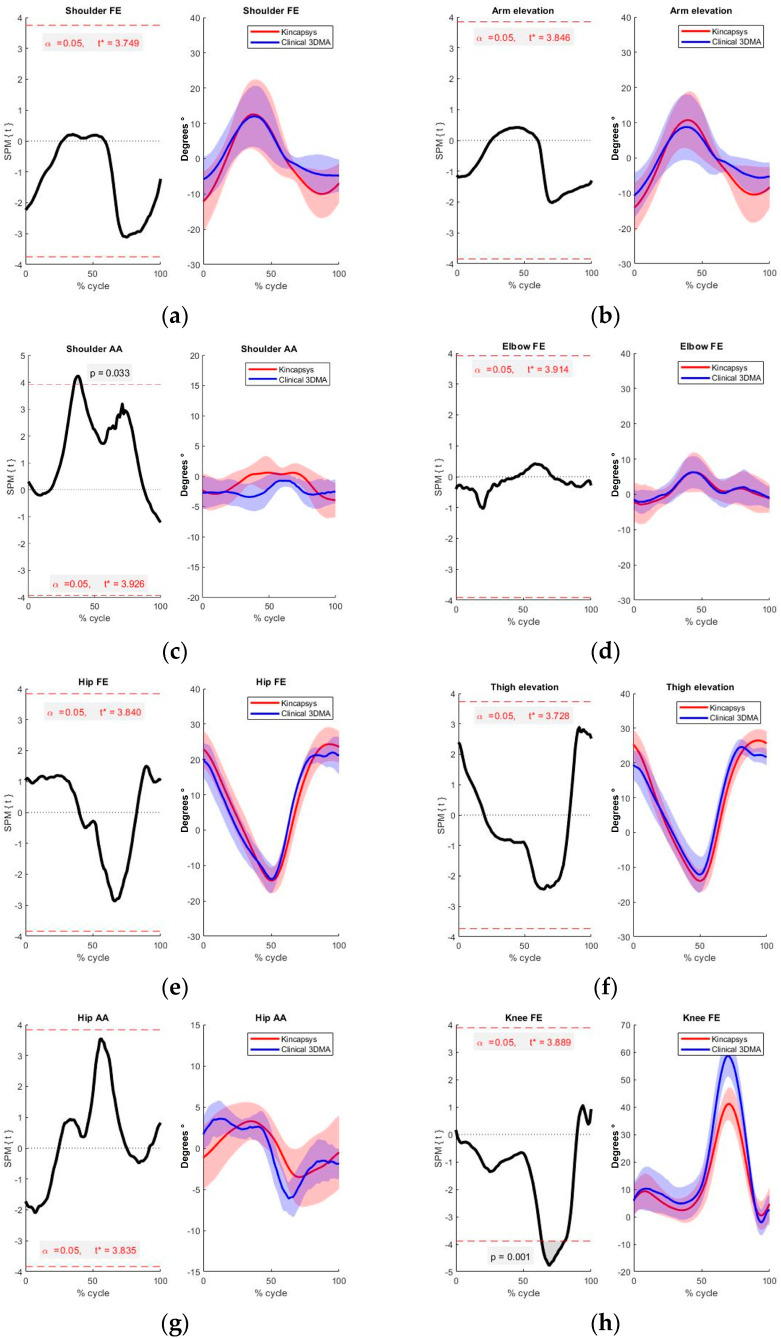
Comparison between systems for the non-pathological sides. On the right side of each figure, the blue curve represents the movement captured by Clinical 3DMA, and the red curve data extracted from Kincapsys. Comparison using SPM 1D is shown on the left side of each figure, the shaded area indicating regions of the curves with the significant difference. (**a**) shoulder flexion-extension, (**b**) arm elevation, (**c**) shoulder abd/add, (**d**) elbow flexion-extension, (**e**) hip flexion-extension, (**f**) thigh elevation, (**g**) hip abd/add, (**h**) knee flexion–extension.

**Figure 7 sensors-24-06351-f007:**
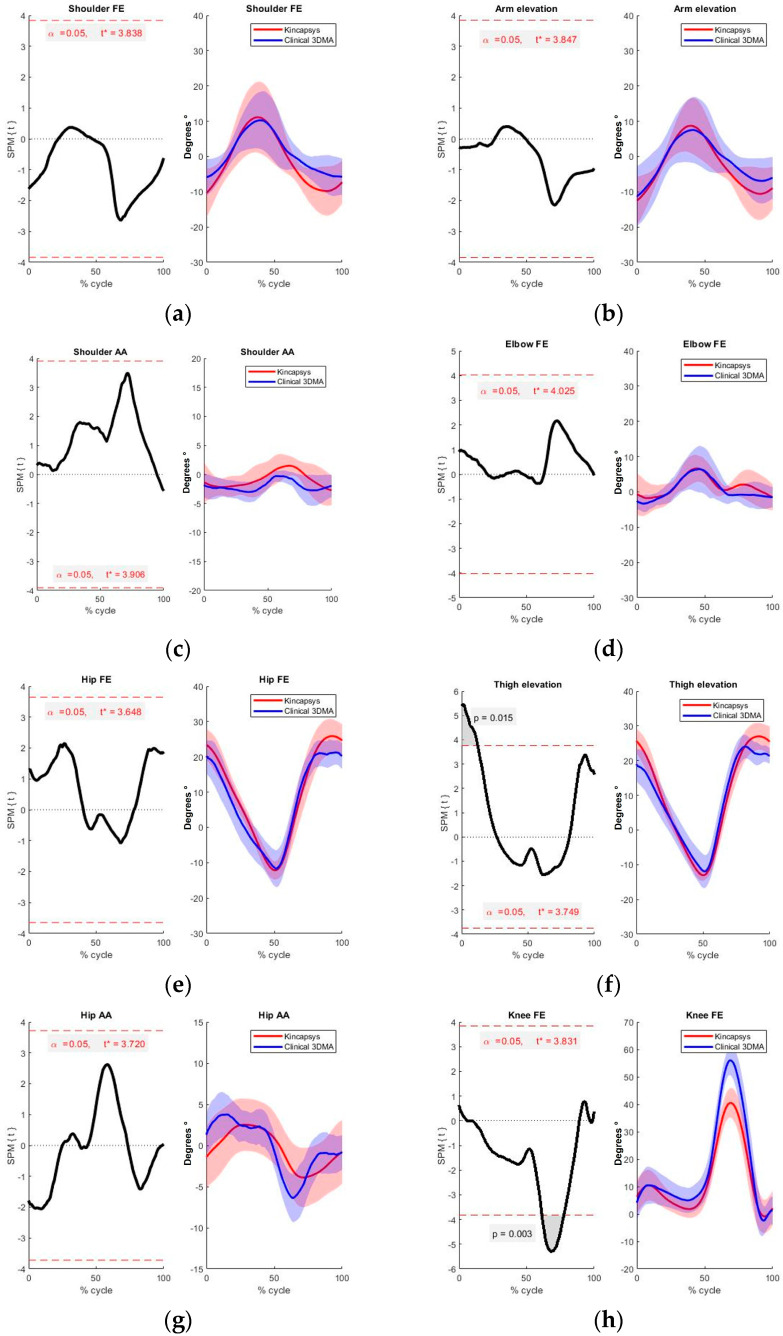
Comparison between systems for the pathological sides. On the right side of each figure, the blue curve represents the movement captured by Clinical 3DMA, and the red curve data extracted from Kincapsys. Comparison using SPM 1D is shown on the left side of each figure, the shaded area indicating regions of the curves with the significant difference. (**a**) shoulder flexion-extension, (**b**) arm elevation, (**c**) shoulder abd/add, (**d**) elbow flexion-extension, (**e**) hip flexion-extension, (**f**) thigh elevation, (**g**) hip ABD/ADD, (**h**) knee flexion-extension.

**Table 1 sensors-24-06351-t001:** Pain levels in volunteers.

Volunteer	Maximum Pain	Average Pain	Test Day Pain
**1**	8	3	3
**2**	6	2	2
**3**	10	3	3
**4**	6	3	4
**5**	7	2	3
**6**	6	4	3
**7**	8	3	2
**8**	7	5	5
**9**	10	7	6
**10**	8	4	4
**Mean**	7.6	3.6	3.5
**Standard deviation**	1.5	1.5	1.3

**Table 2 sensors-24-06351-t002:** Kinect markers used for the development of the motion capture system.

Kinect Markers	Kinect Markers	Kinect Markers
right_shoulder	back_hip	right_ankle
left_shoulder	right_hip	left_ankle
right_elbow	left_hip	right_foot
left_elbow	right_knee	left_foot
right_wrist	left_knee	front_head
front_head	neck	Spine_shoulder
top_head	Spine_mid	
back_head	Spine_base	

**Table 3 sensors-24-06351-t003:** Data obtained from SPM 1D comparisons for movements of the non-pathological side.

Movement	t*	*p*-Value
Shoulder FE	3.75	N/A
Arm elevation	3.85	N/A
Shoulder ABD/ADD	3.93	0.033
Elbow FE	3.91	N/A
Hip FE	3.84	N/A
Thigh elevation	3.73	N/A
Hip ABD/ADD	3.84	N/A
Knee FE	3.89	0.001

**Table 4 sensors-24-06351-t004:** ANOVA RM for Non-pathological side.

		S of Squares	DOF	Mean Square	F-Ratio	*p*-Value
**Shoulder FE**	**Systems**	319.26	1	319.26	4.098	0.074
	**Error**	701.13	9	77.90		
**Arm elevation**	**Systems**	161.21	1	161.20	1.460	0.258
	**Error**	994.10	9	110.46		
**Shoulder AA**	**Systems**	70.42	1	70.42	6.195	0.034
	**Error**	318.57	9	35.40		
**Elbow FE**	**Systems**	5.53	1	5.53	0.098	0.761
	**Error**	506.80	9	56.31		
**Hip FE**	**Systems**	31.95	1	31.95	0.2311	0.642
	**Error**	1244.34	9	138.26		
**Thigh elevation**	**Systems**	0.02	1	0.02	0.0003	0.987
	**Error**	719.40	9	79.93		
**Hip AA**	**Systems**	1.40	1	1.40	0.0294	0.868
	**Error**	429.20	9	47.69		
**Knee FE**	**Systems**	1062.75	1	1062.75	5.3431	0.046
	**Error**	1790.13	9	198.90		

**Table 5 sensors-24-06351-t005:** Tukey test (*p*-values) for shoulder AA and knee FE in non-pathological comparison.

%	Shoulder AA	Knee FE
**0**	1.00	1.00
**10**	1.00	1.00
**20**	1.00	0.95
**30**	0.23	0.99
**40**	5 × 10^−3^	0.98
**50**	0.23	0.98
**60**	0.99	1.7 × 10^−4^
**70**	0.71	1.7 × 10^−4^
**80**	0.75	1.7 × 10^−4^
**90**	1.00	1.00
**100**	0.98	0.98

**Table 6 sensors-24-06351-t006:** Data obtained from SPM 1D comparisons for movements of the pathological side.

Movement	t*	*p*-Value
**Shoulder FE**	3.84	N/A
**Arm elevation**	3.85	N/A
**Shoulder ABD/ADD**	3.90	N/A
**Elbow FE**	4.02	N/A
**Hip FE**	3.65	N/A
**Thigh elevation**	3.75	0.015
**Hip ABD/ADD**	3.72	N/A
**Knee FE**	3.83	0.003

**Table 7 sensors-24-06351-t007:** ANOVA RM for pathological side.

		S of Squares	DOF	Mean Square	F-Ratio	*p*-Value
**Shoulder FE**	**Systems**	235.42	1	235.43	9.15	0.014
	**Error**	231.66	9	25.74		
**Arm elevation**	**Systems**	116.23	1	116.23	1.40	0.266
	**Error**	745.47	9	82.83		
**Shoulder AA**	**Systems**	60.10	1	60.10	2.58	0.143
	**Error**	209.60	9	23.29		
**Elbow FE**	**Systems**	33.42	1	33.43	1.09	0.323
	**Error**	275.74	9	30.64		
**Hip FE**	**Systems**	168.74	1	168.74	0.86	0.378
	**Error**	1769.31	9	196.59		
**Thigh elevation**	**Systems**	70.01	1	70.01	0.80	0.394
	**Error**	786.14	9	87.35		
**Hip AA**	**Systems**	5.59	1	5.59	0.10	0.758
	**Error**	499.73	9	55.53		
**Knee FE**	**Systems**	781.89	1	781.89	10.91	0.009
	**Error**	1790.13	9	198.90		

**Table 8 sensors-24-06351-t008:** Tukey test (*p*-values) for shoulder FE and knee FE in pathological comparison.

%	Shoulder FE	Knee FE
**0**	4.2 × 10^−3^	0.99
**10**	0.88	1.00
**20**	1.00	0.99
**30**	0.99	0.97
**40**	1.00	0.91
**50**	1.00	0.98
**60**	0.99	2.2 × 10^−4^
**70**	5.5 × 10^−3^	1.7 × 10^−4^
**80**	2.8 × 10^−4^	1.9 × 10^−4^
**90**	3.0 × 10^−3^	1.00
**100**	0.99	1.00

**Table 9 sensors-24-06351-t009:** Summary of the relevant statistical results obtained from the comparison between measurement systems.

	Non-Pathological Side			Pathological Side		
	SPM-1D		RM ANOVA	SPM-1D		RM ANOVA
Movement	t*	*p*-Value	*p*-Value—%Gait Cycle	t*	*p*-Value	*p*-Value—%Gait Cycle
**Shoulder FE**	3.75	N/A	0.074	3.84	N/A	0.014 (0%, 70–90%)
**Arm elevation**	3.85	N/A	0.258	3.85	N/A	0.266
**Shoulder A/A**	3.93	0.033	0.034 (40%)	3.90	N/A	0.143
**Elbow FE**	3.91	N/A	0.761	4.02	N/A	0.323
**Hip FE**	3.84	N/A	0.642	3.65	N/A	0.378
**Thigh Elevation**	3.73	N/A	0.987	3.75	0.015	0.394
**Hip A/A**	3.84	N/A	0.868	3.72	N/A	0.758
**Knee FE**	3.89	0.001	0.046 (60–80%)	3.83	0.003	0.009 (60–80%)

## Data Availability

The results obtained constitute an integral component of the doctoral thesis authored by the principal researcher. As these findings are currently in a provisional state, there exists a limitation on the immediate publication of the gathered data until the culmination of the doctoral studies. Subsequent to the conclusion of this academic pursuit, the complete dataset can be obtained directly from the corresponding author upon request.

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
