# Peer review of "Kinect-Based Gait Analysis System Design and Concurrent Validity in Persons with Anterolateral Shoulder Pain Syndrome, Results from a Pilot Study"

_sensors, 2024, doi:10.3390/s24196351_

Round 1
Reviewer 1 Report
Comments and Suggestions for Authors
Dear Authors,
I enclose the pdf file of the paper that contains my comments and which you're expected to carefully address.
Best regards,

Comments on the Quality of English LanguageMinor editing of English language required
Author Response
Response to Reviewer 1 Comments
We sincerely appreciate the valuable time and effort that you have devoted to carefully evaluating the proposed article. Your commitment to the review process is crucial for the ongoing improvement of our work. In response to your constructive feedback, we have made significant adjustments. We value your guidance that enhances the quality and clarity of our work.
Please find the detailed responses below:
- We have adjusted the title as per your suggestion.
- Modification of Figure 1 to avoid copyright conflicts.
- In the text, it is specified that data capture and analysis were carried out under the Declaration of Helsinki and Resolution 8430 of 1993 from the Ministry of Health of Colombia (the country where the study took place).
- Information about the study's environment, location, and participating volunteers has been provided.
- We have highlighted factors that could have influenced the measurements, along with recommendations taken to minimize these effects.
- It is noted that patient diagnosis was conducted through the Colombian healthcare system. However, due to data treatment policies, the names of the professionals were not requested. Additionally, it is important to clarify that patient evolution was not assessed during the study.
- We have provided a more in-depth explanation of the statistical methodologies used. Nevertheless, for a deeper understanding, we recommend reviewing the authors' references.
- A paragraph has been added outlining the study's limitations, particularly the small sample population and the reasons behind this choice.
- Another paragraph has been added indicating the estimated dollar value of the system in comparison to traditional commercial systems.
- A paragraph was added clarifying that although suggestions for orthopedic rehabilitation cannot be given based on the study carried out, the equipment implemented does allow for follow-up of rehabilitation therapies, given its low cost and easy use.
- Other minor adjustments within the document.
We hope these modifications meet your expectations, and we would be delighted to receive any additional feedback you may have.
However, since the study did not aim to assess the evolution or behavior of the shoulder injury, it is not possible to provide detailed information on the following aspects:
- Description of damages.
- Treatment compliance (acceptability).
- Handling of missing data.
Additionally, the sample calculation can be expressed by the authors, yielding an expected total of 32 volunteers. However, during the study period between 2020 and 2021, the availability of volunteers drastically decreased, largely due to the COVID-19 pandemic and measures implemented by the national government of Colombia, significantly restricting people's mobility. This resulted in a maximum of only 10 available individuals. Therefore, future work is proposed to involve measurements with a larger number of volunteers, including individuals without orthopedic alterations, to obtain statistically more accurate comparisons and, additionally, to obtain normality parameters with this equipment.
The changes made to the document are highlighted.
Any additional comments will be warmly welcomed by the authors.
Sincerely,
Ing. Fredy David Bernal Castillo M.Sc.
Reviewer 2 Report
Comments and Suggestions for Authors
The concept and outcomes of this study are intriguing and beneficial. A notable strength of this research lies in its demonstration of Kinect's potential for assessing physical disabilities related to specific health problems. However, there are several aspects that the authors need to address:
-
The sample size of test subjects is quite limited, which could affect the generalizability of the findings.
-
The rationale behind the extensive use of statistical tests requires clearer explanation. It would be advantageous if the authors could elucidate the framework of their statistical analyses, summarizing all test results to present a comprehensive overview of their findings.
-
With the advent of many AI algorithms capable of detecting human body joints through images taken by standard cameras and mobile phones, the authors should discuss the specific need for employing the Kinect system in their research. This discussion would highlight Kinect's unique advantages or the specific reasons for its use in this context.
Author Response
Response to Reviewer 2 Comments
Thank you very much for the time and dedication to reviewing this research work; contributions from experts in the field are always welcomed to enhance the quality of the undertaken projects. The changes made to the document based on your feedback can be found below:
- The authors recognize the limited sample size and are aware of the limitations stemming from it, with the sample calculation projecting an expected total of 32 volunteers. However, during the study period from 2020 to 2021, the availability of volunteers experienced a notable decline, primarily attributed to the circumstances detailed in the document, including the impact of the COVID-19 pandemic and measures implemented by the national government of Colombia, significantly restricting people's mobility. As a result, the maximum number of individuals available was limited to just 10. Consequently, future efforts contemplate conducting measurements with a larger number of volunteers, encompassing individuals without orthopedic alterations. The goal is to achieve statistically more precise comparisons and, additionally, obtain normality parameters using this equipment.
- A more in-depth explanation of the statistical methodologies employed was provided, and a table summarizing the relevant data found in the study was included at the end of the results section.
- In the discussion section, the points raised by the reviewer were addressed, drawing comparisons with other joint measurement technologies, such as those utilizing smartphone cameras. In the same paragraph, the advantages of using systems like Kinect, particularly its depth sensor for measuring these movements, were emphasized.
- Modification of Figure 1 to avoid copyright conflicts.
- Other minor adjustments within the document.
The changes made to the document are highlighted.
Any additional comments will be warmly welcomed by the authors.
Sincerely,
Ing. Fredy David Bernal Castillo M.Sc.
Round 2
Reviewer 1 Report
Comments and Suggestions for Authors
Dear Authors,
thanks of addressing my comments and for giving a complete explanation to my queries. I'd further suggest to add to the Results section the sentence you already provided with the reply:
"Since the study did not aim to assess the evolution or behavior of the shoulder injury, it is not possible to provide detailed information on the following aspects: description of damages; treatment compliance (acceptability); handling of missing data."
Best regards,
Author Response
I appreciate all the observations and suggestions made to the document. We have already added the suggested paragraph to the end of the results section (between lines 315 and 317 of the document). Any additional comments will be welcomed. Regards.
